# Sensitivity analysis of shock distributions in the world economy

**Viktor Domazetoski**[1]*, **Maryan Rizinski**[2,3], **Dimitar Trajanov**[2,3], **Ljupco Kocarev**[1,2]

**1** Research Center for Computer Science and Information Technologies, Macedonian Academy of Sciences and Arts, Skopje, Macedonia, **2** Faculty of Computer Science and Engineering, Ss. Cyril and Methodius University, Skopje, Macedonia, **3** Department of Computer Science, Metropolitan College, Boston University, Boston, MA, United States of America

* v.domazetoski@manu.edu.mk

**Data Availability Statement:** All Data are from a third party. World Input-Output Data is available from the 2016 release of the World-Input-Output Database (https://www.rug.nl/ggdc/valuechain/wiod/wiod-2016-release). Total factor productivity

## Abstract

With the ever increasing interconnectedness among countries and industries, globalization has empowered economies and promoted international trade, capital flow and labor mobility, leading to improved products and services. However, the growing interdependence has also propelled an inherent reliance on joint cooperation which has considerably influenced the complexity of global value chains (GVCs). This plays a significant role in policy decisions, raising questions about trade risks that originate from such interdependence. In this paper, we study the impact of network linkage disturbances on the output supply and input demand of countries. We model the network interconnectedness of countries according to the latest 2016 release of the World Input–Output Database (WIOD) that includes data tables for the period 2000-2014 covering 43 countries as well as a model for the Rest of the World (ROW). We assess the shock distributions across the world economy by quantifying the changes in the network linkages using sensitivity analysis. Our contribution is in the definition of a shock tensor with the purpose of evaluating the impact of link sensitivity. The shock tensor is a straightforward yet comprehensive tool that allows us to obtain ample results at various levels of granularity when combining it with aggregation operators. Our study introduces a novel methodology that enables us to acquire input and output link sensitivities for all country pairings when an economic shock initiates or concludes within a country of interest. This innovative approach also facilitates the analysis of evolving trends in these link sensitivities, providing a comprehensive understanding of the dynamics of shock propagation across the global network. Taking advantage of the time-series nature of the WIOD, our results reveal illustrative visualizations and quantative measures that characterize patterns of shock distribution and relationships among countries throughout the period from 2000 to 2014. Our methodology and results not only uncover valuable trends but also establish a structured approach to better understand the aggregate effects of shock distributions. Thus, this study could be helpful for policy makers to assess trade relationships between countries and obtain quantitative insights for making informed decisions as well as explore the overall state of the globalization as a whole.

data is available from the April, 2022 release of the Conference Board Total Economy Database™ (https://www.conference-board.org/data/economydatabase/total-economy-database-productivity).

**Funding:** The authors received no specific funding for this work.

**Competing interests:** The authors have declared that no competing interests exist.

## Introduction

Sensitivity analysis of a country Input-Output (IO) models has a long history and has been proceeded along three paths. The first path considers the extent to which errors interact through matrix inversion to bias the results and here we mention some results in this direction: Simonovitz [1] discusses underestimation and overestimation of the Leontief Inverse, while Lahiri and Satchel [2] derive necessary and sufficient conditions for the over- and underestimation of inverse elements, assuming that prices are the source of the stochastic errors. The second path builds on the results of Sherman and Morrison [3] who traced the effects of a discrete change in a single technical coefficient. The third path explores the feasibility of using Monte Carlo simulation to study stochastic error propagation through IO models, which has been pioneered by Clark W. Bullard and Anthony V. Sebald in a series of papers, see, for example [4, 5].

As an essential factor for growth across the world, globalization has not only helped industrialized countries to expand their exports but also provided opportunities to developing nations to diversify their economies [6]. This widespread and ongoing economic growth due to globalization has been accompanied by the rise of international competition, reshaping global production, trade and organization of industries [7, 8]. The competition landscape has been profoundly changed as a consequence of the liberalization and deregulation of international trade and investments as well as the development and wide adoption of information and communication technologies (ICT); such competitive circumstances led to sophisticated patterns of globally distributed economic activity [9]. Enabled by technological advancements, the global economic activity and production processes have naturally become more geographically fragmented which has resulted in the emergence of global value chains (GVCs). On the other hand, increased trade has been fostered by the involvement of more industries and countries in the value chains as well as the distribution of goods creation throughout different parts of the world [10]. To put it another way, just as globalization has changed the essence and scope of international competition, the competitive environment has become the driver for intensified trade which exhibits global expansion, fragmentation and structural complexity.

The study in [7] points out that *"international competition is best understood by looking at the global organization of industries and how countries rise and fall within these industries"*. Understanding how global industries organize and reconfigure is in the main focus of the GVC research; the GVC framework focuses on value creation and capture across diverse activities and products (goods and services) while emphasizing expansion and fragmentation of production networks [7]. Going through different phases of production, from supply of raw materials through assembly of goods to their delivery and consumption, trade dynamics among countries and industries within GVCs can be inherently modeled as networks thereby illustrating that the global economy is increasingly integrated, interdependent and specialized [11]. Major international organizations like the World Bank, the World Trade Organization, the International Labor Organization, and the U.S. Agency for International Development have utilized GVCs in their research and policy development [7].

With the widespread interconnectedness of the global economy, it becomes increasingly important to study the impact of demand and supply shocks and especially their propagation throughout the network. Shocks can negatively affect both suppliers and consumers: production declines or collapse of demand are an equal threat to supply-chain continuity and sustainability. Reference [11] draws attention that *"trade openness is a double-edged sword"* as it can insulate economies against domestic and regional shocks but can also leave them vulnerable by increasing the susceptibility to external shocks. From estimating supply or demand slowdown to assessing recovery capabilities, researchers and policy makers are interested in

developing risk assessment strategies to analyze and mitigate the adverse consequences of supply chain disruptions. However, while the yearly trade volume among countries and industries has been known and well documented using various databases, there is yet work to be done in the literature to examine models that measure shock propagation from a sensitivity perspective, i.e. the extent to which shocks originating from a country or industry affect other countries or industries.

Sensitivity analysis of World-Input-Output models and/or multi-regional input–output (MRIO) models has also recently attracted great deal of interest due to the emergence of GVCs. To assess the impact of shocks in a world of global value chains, several models and approaches have been developed. Here we mention several, thus, for example, Caliendo and Parro [12] build a Ricardian model with sectoral linkages and trade in intermediate goods to quantify the welfare effects from tariff changes, while [13] offers a structural gravity approach to quantify output and welfare effects. The paper [14] uses the sectoral World Input–Output Database (WIOD) to evaluate the impact in terms of value added and employment of different scenarios of Brexit for 56 industries in the 27 Member States of the European Union, as well as the United Kingdom. Gerschel, Martinez & Mejean, [15] employ the WIOD dataset to measure the share of a particular sector in a particular country as a source in the gross output of a particular sector of a given country. Using the Leontief inverse matrix, the authors measure how the gross output of each sector from each country is exposed to shocks affecting China, both directly and indirectly. Using the World Input-Output Database (WIOD), Mandel and Veetil [16] study the effects of national lockdowns on global GDP in a non-equilibrium framework. Pichler and Farmer [17] combine WIOD framework and the approach of del Rio-Chanona et al. [18] to compute supply shocks for every industry during a lockdown of Covid-19 in Germany, Italy, and Spain.

This paper builds on the prior research in [19] where discrete-time absorbing Markov chains are proposed to model the structure and interdependence among country-industry pairs of the world economy. Several novel properties are designed based on the discrete-time absorbing Markov chains approach with the aim of evaluating the volatility and risk when shaping production chain lengths. In addition, the study shows that the input and output chains exhibit exactly the same quasi-stationary product distribution, meaning that the time spent in a state before absorption is invariant to the changes of the network type [19]. The paper also suggests several global metrics, including the probability distribution of global value added/final output, provide guidance for policy makers when estimating the resilience of world trading system and forecasting the macroeconomic developments. Most macroeconomic models typically derive from the Leontief's classical work on input-output tables that characterize global production networks [20]. They are characterized with multi-regional input–output (MRIO) models and there are several independently constructed global MRIO databases such as the WIOD used in this paper in combination with numerical computations [21]. While the production network can be observed as a medium for propagating shocks throughout the economy, the risks of substantial aggregate fluctuations due to shocks is abandoned in the literature due to the *"diversification argument"* [19, 22]. The 'diversification argument' posits that within an economy comprised of $n$ industries experiencing independent shocks, the magnitude of aggregate fluctuations would roughly scale inversely with the square root of $n$. This suggests that when examining highly detailed or disaggregated levels of the economy, individual shocks tend to have relatively minor effects on overall fluctuations. However, the diversification argument has several limiting assumptions such as the independence of shocks and industry homogeneity. Notably, it also takes no account of linkages between industries that can also serve as a channel for shock propagation through the network [23–28].

In this paper, we tackle that gap in the literature by exploring the influence of linkages between countries and industries on sensitivity of shock propagation. For that purpose, we model the network interconnectedness of countries based on the 2016 release of the World Input–Output Database (WIOD). The WIOD contains data for the period 2000–2014 and includes 43 countries as well as a model for the Rest of the World (ROW). We examine the characteristics of the network vis-à-vis the propagation of shocks in the world economy. To assess the impact of shock distributions, we focus on the changes in the network linkages using sensitivity analysis. We define a shock tensor with the purpose of evaluating the impact of link sensitivity. Using the time-series WIOD data in combination with the shock tensor, we obtain results and visualizations that reveal patterns of shock distribution and relationships among countries throughout the period from 2000 to 2014. The results show trends about the aggregate effects of shock distributions which could be helpful to policy makers in assessing risks arising from country or industry interdependence and trade relationships.

The rest of the paper is organized as follows. The section on "Materials and Methods" describes the data organization and network aggregation aspects, introducing preliminaries that serve as a foundation for the remainder of the study. This section also outlines the theoretical model underlying our analysis of network linkage disturbances and introduces the shock tensor including means for its calculation. For completeness purposes, detailed proofs of the theorems are given in the appendices. The section on "Results and Discussion" intertwines ample visualization diagrams with a discussion on sensitivity-wise findings obtained by different aggregations of the shock tensor. The last section "Conclusion" presents concluding remarks of the research.

## Materials and methods

### Data

**World Input-Output Database.**   The World Input-Output Database (WIOD) [21] contains annual time-series of World Input-Output Tables (WIOT) describing the trade relationships (sales and purchases) between producers and consumers both within an individual economy (country) as well as between different economies as expressed by their respective bilateral trade data. The latest 2016 release of the WIOD, which we will use within this work, incorporates a total of 43 economies, including 28 EU member states (as of 2014) and 15 other major economies, accounting for more than 85% of the world GDP. Furthermore, it includes a residual region with data about the remaining countries of the global economy, named as *Rest of World* and abbreviated as "ROW". Spanning a 15-year period, from 2000 to 2014, the WIOD classifies trade data into 56 sectors (industries) according to the International Standard Industrial Classification Revision 4 (ISIC Rev. 4) which is the United Nation's international reference classification of productive activities.

**Total Economy Database.**   To calculate the Hicks-neutral productivity shocks we use the April, 2022 release of the Conference Board Total Economy Database™ [29]. We start with a baseline of 100 for each country for the year 1990. Then, using the Total Factor Productivity data we calculate the productivity shocks throughout time for each country. These values are then logarithmically transformed, standardized to a unit variance and normalized to a sum of one due to the model specifications outlined below.

**Data organization.**   A tensor is a multidimensional object which is a generalized version of a vector. The dimensionality of this object is called the order of the tensor. Within the paper we will use different symbols to indicate the order of the tensor. As such, scalars (tensors of order zero) are illustrated by lowercase or uppercase letters, e.g. x or X, while vectors (tensors of order one) will be denoted by a boldface lowercase letters, e.g., **x**. Matrices (tensors of order

two) will be denoted by a boldface uppercase letter, e.g., $\mathbf{X}$. Tensors of a higher order will be symbolized by a math calligraphy letter, e.g. $\mathcal{X}$.

The information within the WIOT is organized as the following two tensors: a 4-order tensor $\mathcal{Z} \in \mathbb{R}^{J \times J \times S \times S}$, and a 3-order tensor $\mathcal{F} \in \mathbb{R}^{J \times J \times S}$ where $J$ and $S$ denote the number of countries and industries, respectively. The entries of $\mathcal{Z}$ describe the intermediate purchases (input flows) $z_{ij}^{rs}$ by industry $s$ in country $j$ from sector $r$ in country $i$. The entries $f_{ij}^{r}$ denote the final use in each country $j$ of output originating from sector $r$ in country $i$.

Given the tensors $\mathcal{Z}$ and $\mathcal{F}$, we can calculate the matrices $\mathbf{F}$, $\mathbf{X}$ and $\mathbf{B}$ as follows:

$$f_i^r = \sum_{j=1}^{J} f_{ij}^r, \tag{1}$$

$$x_i^r = \sum_{s=1}^{S} \sum_{j=1}^{J} z_{ij}^{rs} + f_i^r, \tag{2}$$

$$\beta_i^r = \frac{f_i^r}{\sum_{i,r} f_i^r} \tag{3}$$

where the element $f_i^r$ stands for the value of output from sector $r$ in country $i$ intended for final consumers worldwide, $x_i^r$ represents the value of gross output originating from sector $r$ in country $i$ and $\beta_i^r$ represents the weight of the good produced by the country-industry pair $(i, r)$ calculated by normalizing $f_i^r$.

**World-input network.** *We can now define the World-input network, which is represented by the $(J \times S) \times (J \times S)$ adjacency matrix $\mathbf{A}^{(1)} = [a_{\hat{i}\hat{j}}^{(1)}]$ where $\hat{i}$ and $\hat{j}$ represent industry pairs $(i, r)$ and $(j, s)$ respectively, such that $a_{\hat{i}\hat{j}}^{(1)} = z_{ij}^{rs}/x_j^s$. In this network, the country-industry pair $(i, r)$ sells intermediate inputs to other country-industry pairs $(j, s)$'s in the world economy. From the World-input network, we can further define aggregations. In the Country World-input network we sum over the sectors resulting in the $J \times J$ adjacency matrix which reads*:

$$a_{ij}^{(2)} = \frac{\sum_{r,s} z_{ij}^{rs}}{\sum_s x_j^s} \tag{4}$$

*Equivalently, in the Sector World-input network we sum over the countries resulting in the $S \times S$ adjacency matrix which reads*:

$$a_{rs}^{(3)} = \frac{\sum_{i,j} z_{ij}^{rs}}{\sum_s x_j^s} \tag{5}$$

*Within this paper we work with the Country world-input network $\mathbf{A}^{(2)}$, however the introduced methodology stands for all three variations of the World-input network and will be explored upon in future work.*

## Theoretical model

Let $\mathbf{A} = [a_{ij}]$ be either $\mathbf{A}^{(1)}$, $\mathbf{A}^{(2)}$, or $\mathbf{A}^{(3)}$ matrix. We assume that $i, j = 1, \ldots, n$, $a_{ij} \geq 0$ and $\sum_j a_{ij} < 1$, so that the largest eigenvalue of $\mathbf{A}$ is positive and less than 1.

**Remarks and assumptions.** The model we consider in this study is detailed mathematically in S1 Appendix and assumes the following three assumptions: (i) Cobb-Douglas preferences and technologies, (ii) a single factor (labor) of production, and (iii) constant returns to

scale. These assumptions imply that the graph $G$ represents a world-input network and that the productivity shocks propagate "downstream" from one industry to its customers, its customers' customers, and so on.

**Aggregate effects of network linkage disturbances.** Our analysis focuses on the macro-economic impact of the network linkage disturbances by examining the relationship between the changes of the aggregate output and the changes of network linkages. Assuming that $a_{ij}$ are endogenous variables, we analyse the impact of a change in the network linkage $a_{ij}$ on the Leontief-inverse matrix $\mathbf{L} = (\mathbf{I} - \mathbf{A})^{-1}$ [30–32]. The analysis adequately answers certain counterfactual questions about the network linkages in our model. The change in the outcome $X$, in response to a change in a parameter $\theta$ (network linkage), can be approached via a quantity called *sensitivity* defined as $\frac{dX}{d\theta}$ or a quantity called *elasticity* defined as $\frac{d \log X}{d \log \theta}$. In the case of network linkages, it is more reasonable to use, due to the fact that we are measuring the absolute change in the Leontief-inverse matrix instead of measuring a proportional change which is more suited for a elasticity analysis. The vec operator transforms an $n \times n$ matrix $\mathbf{A}$ into an $n^2$-dimensional vector: $vec\, \mathbf{A} = [a_{11}, a_{21}, \ldots, a_{n1}, a_{12}, a_{22}, \ldots, a_{nn}]^T$. Let $\mathbf{x}$ and $\mathbf{y}$ be $n \times 1$ vectors. The derivative of $\mathbf{y}$ with respect to $\mathbf{x}$ is defined to be the $n \times n$ matrix whose $(i, j)$ entry is derivative of $y_i$ with respect to $x_j$. The derivative of the $n \times n$ matrix $\mathbf{Y}$ with respect to the $n \times n$ matrix $\mathbf{X}$ is the $n^2 \times n^2$ matrix defined as follows:

$$\frac{d \operatorname{vec} \mathbf{Y}}{d(\operatorname{vec} \mathbf{X})^T}$$

**Theorem 1** *Let* $\mathbf{L} = [\ell_{ij}]$. *(i) The derivative of the Leontief-inverse matrix $\mathbf{L}$ with respect to the scalar $a_{ij}$ is an $n \times n$ matrix:*

$$\mathbf{L}_{ij} \equiv \frac{d\mathbf{L}}{da_{ij}} = \left[\ell_{jp}\ell_{qi}\right], \tag{6}$$

*where $i, j$ are fixed and $p, q = 1, \ldots, n$.*

*(ii) The derivative of the Leontief-inverse matrix $\mathbf{L}$ with respect to the matrix $\mathbf{A}$ is an $n^2 \times n^2$ matrix:*

$$\frac{d\mathbf{L}}{d\mathbf{A}} = \begin{bmatrix} \ell_{11}\ell_{11} & \ell_{21}\ell_{21} & \cdots & \ell_{n1}\ell_{nn} \\ \ell_{12}\ell_{11} & \ell_{22}\ell_{21} & \cdots & \ell_{n2}\ell_{nn} \\ \vdots & \vdots & \ddots & \vdots \\ \ell_{1n}\ell_{n1} & \ell_{2n}\ell_{n1} & \cdots & \ell_{nn}\ell_{nn} \end{bmatrix} \tag{7}$$

**Shock tensor.** The celebrated theorem of [33] states that for efficient economies and under minimal assumptions, the first-order macroeconomic impact of microeconomic shocks is given by (see, for example, [25]):

$$\frac{d \log Y}{d \log \zeta_i} = \lambda_i \tag{8}$$

where $Y$ is the equilibrium aggregate output, $\zeta_i$ is Hicks-neutral technology, and $\lambda_i$ is Domar weight. We consider the model suggested in [23] (see also S1 Appendix), for which it follows

that:

$$\log(\text{GWP}) \quad = \quad \sum_{i=1}^{n} \varepsilon_i \lambda_i \tag{9}$$

$$= \quad \sum_{i=1}^{n} \varepsilon_i \sum_{j=1}^{n} \ell_{ij} \beta_j \tag{10}$$

where GWP is the Gross World Product as defined in [23], $\varepsilon_i = \log A_i$ and $\ell_{ij}$ is the $(i, j)$ entry of the Leontief-inverse matrix, $\mathbf{L} = (\mathbf{I} - \mathbf{A})^{-1} \equiv [\ell_{ij}]$, while $\beta_i \in (0, 1)$ designates the weight of the good $i$ (produced by country-industry pair $i$) in the representative household's preferences (with the normalization $\Sigma_i \beta_i = 1$). Eq (9) shows that in a competitive world economy with constant returns to scale technologies, the world aggregate output is a linear combination of country-industry level productivity shocks, with coefficients $\lambda_i$ given by Domar weights. Moreover, as per Eq (10), the Domar weight of each country-industry pair $i$ depends only on the preference shares, $\beta_1, \ldots, \beta_n$, and the corresponding column of the world-economy's Leontief-inverse matrix.

For a fixed pair $(u, v)$, let $\Delta a^{uv}$ be the disturbance of the $(u, v)$ entry of the matrix $\mathbf{A}$. This disturbance generates a change in the GWP which we assume can be written as

$$\text{GWP}^{new} = p^{uv}\text{GWP}$$

where $p^{uv}$ is a multiplier of the GWP; $p^{uv}$ can be expressed as a product of the multipliers $p_{ij}^{uv}$ for each link $(i, j)$, that is:

$$p^{uv} = \prod_{i,j} p_{ij}^{uv} \tag{11}$$

Assuming no changes in $\varepsilon_i$ and $\beta_j$, by combining Eqs (9), (10) and (11), it is easy to verify that:

$$\log(\text{GWP}^{new}) = \sum_{i=1}^{n} \varepsilon_i \sum_{j=1}^{n} (\ell_{ij} + \Delta\ell_{ij}^{uv}) \beta_j \tag{12}$$

where $\Delta\ell_{ij}^{uv}$ given as:

$$\Delta\ell_{ij}^{uv} = \frac{\log p_{ij}^{uv}}{\beta_j \varepsilon_i} \tag{13}$$

We then define a tensor $\mathcal{P} \in \mathbb{R}^{J \times J \times J \times J}$, which will be called *shock tensor*, as follows:

$$\mathcal{P} = [p_{ij}^{uv}] \tag{14}$$

The next theorem shows the effect of the disturbance $\Delta a^{uv}$ on the GWP (or GDP).

**Theorem 2** *The tensor $\mathcal{P}$ can be computed as*

$$\mathcal{P} = [\ell_{vi} \ell_{ju} \Delta a^{uv} \beta_j \varepsilon_i], \tag{15}$$

*where $u$ and $v$ denote the indices on the U an V axes where the disturbance originates from, while $i$ and $j$ denote the indices on the I and J axes which consume that disturbance.*

**Model implementation.** We implemented the model in Python 3.9 and the implementation code and user guidelines can be found at the corresponding GitHub repository (github. com/ViktorDomazetoski/Sensitivity-Analysis-of-Shock-Distributions). The code can be used

to analyze the data in more detail, such as to explore the sensitivity matrices of different countries or the trends between country-country link sensitivity. The $\Delta A$ matrix was calculated as a constant increase of 0.01 in each link $a^{uv}$. We focused on six economies of interest: China, USA, Germany, Russia, Japan, Rest of World with a focus on the changes between the years 2000 and 2014.

## Results and discussion

The shock tensor $\mathcal{P}$ shown on Fig 1 allows us to analyze several results depending on the level of detail we are interested in. It should be noted that the values of all heatmap visualizations in the paper are transformed twice logarithmically to further see the details of the matrices which are otherwise dominated by a few extremely large values and that the same color scale is used within each figure to allow for a temporal comparison of the results.

We define two operations to achieve the tensor representation of the shock tensor $\mathcal{P}$. The first operation is introduced using the aggregation operator (:) which represents a cumulative operation along the specified axis such as the calculation of the mean or sum of the elements on that axis. In our study, unless specified otherwise, the aggregation operator uses the geometric mean operation due to the nature of the shock tensor $\mathcal{P}$ which is defined as a multiplicative change in Gross World Product (GWP). An example of using this operator is $\mathcal{P}_{::}^{::}$ which represents a geometric mean across all four axes of the shock tensor. Applying such an operator allows us to get a $(1 \times 1)$ index which can be analyzed throughout time to uncover any temporal trends in the sensitivity of the World Input Output Network. The second operation consists of fixing the value of a specific index, e.g. $i$ to a value $\hat{i}$. An example result of this operation is $\mathcal{P}_{ij}^{\hat{u}\hat{v}}$ meaning that we set the indices of the network disturbance to specific values. For example, if we set $\hat{u} = DEU$ and $\hat{v} = USA$ to represent the disturbance in the country link between Germany (Input) and USA (Output), that will result in a $J \times J$ matrix showing the effect of this disturbance on every other pair of countries within the network.

In the following sections, we will show and analyze various results that can be obtained from the shock tensor $\mathcal{P}$ by applying these two operators.

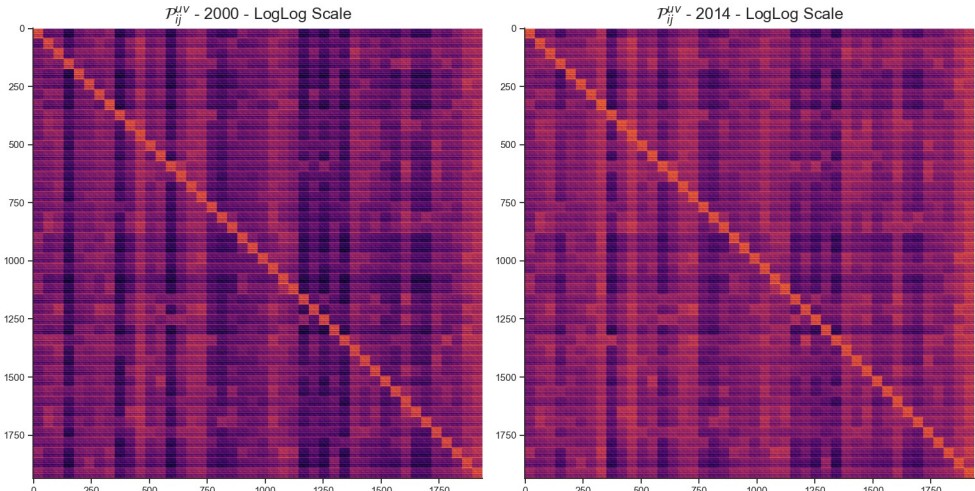

**Fig 1. Heatmap visualization of the shock tensor $\mathcal{P}$ for the years 2000 (A) and 2014 (B).** Each column of the matrix represents a derivative of the corresponding link of the Leontief matrix.

## Disturbance start matrix sensitivity $\mathcal{P}_{::}^{uv}$

Aggregation on the $I$ and $J$ axis of the $\mathcal{P}$ tensor, denoted as $\mathcal{P}_{::}^{uv}$, allows us to get an understanding about the sensitivity of each link. With this approach, if a specific link with indices $u$ and $v$ exhibits a change, we can quantify the impact of that change on the overall world economy. The aggregation $\mathcal{P}_{::}^{uv}$ can be calculated as follows:

$$\mathcal{P}_{::}^{uv} = \left[ \sqrt[J^2]{\prod_{i,j} p_{ij}^{uv}} \right] \tag{16}$$

and results in an $J \times J$ matrix shown in Fig 2. The matrix is characterized by row-wise and column-wise patterns. The darker color visualize lower impact (i.e. lower aggregation) while the brighter colors visualize higher impact (i.e. higher aggregation). The highest values in 2000 can be noticed in the links USA-USA and ROW-USA, while the highest values in 2014 can be noticed in the links China-China and ROW-China. We can see an overall increase in the mean sensitivity and heterogeneity in the matrix between 2000 and 2014 with a few exceptions such as Japan.

## Disturbance end matrix sensitivity $\mathcal{P}_{ij}^{::}$

Aggregation on the $U$ and $V$ axis of the $\mathcal{P}$ tensor, denoted as $\mathcal{P}_{ij}^{::}$, provides us with another approach to analyzing the world economy from a sensitivity perspective. Unlike $\mathcal{P}_{::}^{uv}$ which focuses on understanding the contribution of changes in individual links on the world economy, $\mathcal{P}_{ij}^{::}$ is focused in the opposite direction. In particular, $\mathcal{P}_{ij}^{::}$ is helpful in understanding how much an average change in the overall economy will impact a specific link with indices $i$ and $j$. The aggregation $\mathcal{P}_{ij}^{::}$ can be calculated as follows:

$$\mathcal{P}_{ij}^{::} = \left[ \sqrt[J^2]{\prod_{u,v} p_{ij}^{uv}} \right] \tag{17}$$

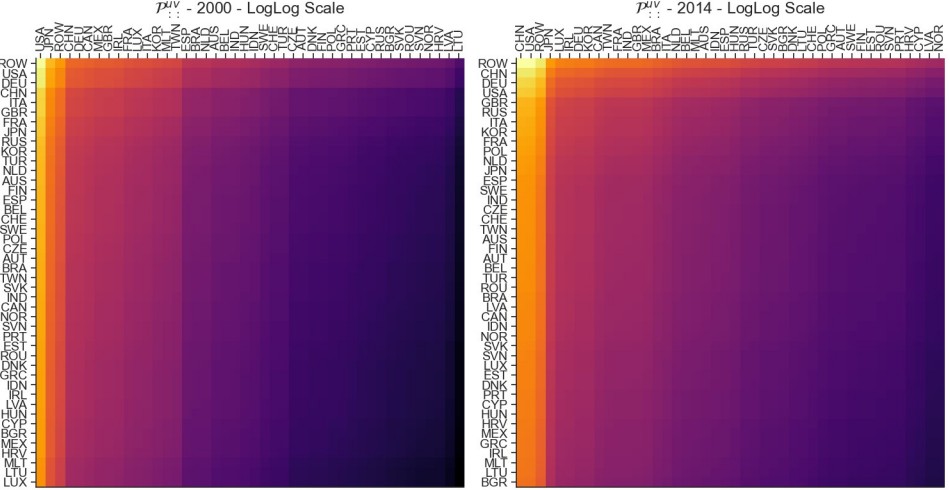

**Fig 2. Heatmap visualization of the $\mathcal{P}_{::}^{uv}$ shock tensor aggregation for the years 2000 (A) and 2014 (B).** This matrix allows us to understand the contribution of changes in individual links on the entire world economy.

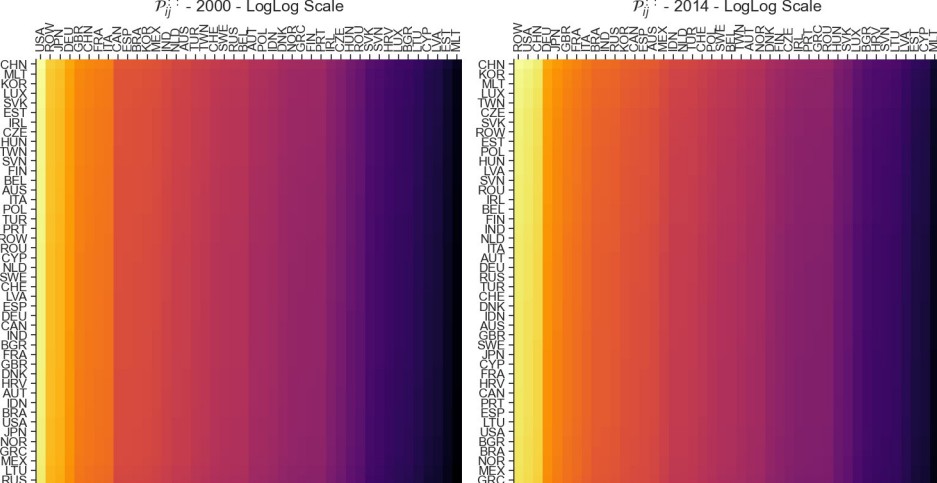

**Fig 3. Heatmap visualization of the $\mathcal{P}_{ij}^{\cdot\cdot}$ shock tensor aggregation for the years 2000 (A) and 2014 (B).** This matrix allows us to understand how an average change in the world economy impacts individual links.

and results in an $J \times J$ matrix shown in Fig 3. Here we can see that most of the heterogeneity is column-wise, denoting that countries are the output in the network.

## Disturbance end fixed matrix sensitivity $\mathcal{P}_{ij}^{\hat{u}\hat{v}}$

If we fix the starting values of the disturbance $u$ and $v$, we will get a resulting matrix that shows how that disturbance will affect the entire economy. Here we are interested in the links where the input and output correspond to the same country which can be obtained $u = v$. We look into several countries such as China, Germany, Japan, Russia and USA as well as the ROW region for the year 2014 and then visualize how the shock propagates in Fig 4. On Fig 4, we can see that for Germany and ROW the shock is propagated more considerably throughout the network when compared with Russia and Japan where the shock is concentrated around the originating country. To quantify this, we can calculate the percentage of the shock contained within a subset of the matrix by normalizing $log(p_{ij}^{\hat{u}\hat{v}})$ by the sum of the matrix. We do this calculation for the sensitivity within the country $p_{uu}^{\hat{u}\hat{u}})$, where the country acts as an exporter $p_{uj}^{\hat{u}\hat{u}})$ for $j$ in $1, \ldots, n$ and $j \neq u$, where the country acts as an importer $p_{iu}^{\hat{u}\hat{u}})$ for $i$ in $1, \ldots, n$ and $i \neq u$. Additionally, we look at top 10 and 22 upper left values of the matrices shown in Fig 4. The results are presented in Table 1. Now we can further see how 62.3% of the shock which originates within Japan ends in Japan, while this is at 26.4% and 32.6% for ROW and Germany. If we look at the input percentages, we see how much of the shock flows into countries which import from the fixed country, with Russia's trade partners being the most affected. Similarly, the output percentages show what percentage of the shock is distributed to countries which act as exporters in the scenario. These values are much higher than the input sensitivities across all countries, with highest values for the Rest of World Model. The top 10 and 22 percentages show us the overall distribution of the shock. Again, Japan and Russia contain 91.8% (96.2%) and 85.0% (94.1%) for the top 10 (top 22) percentages, showing a less global impact on the economy, while for ROW these values are 55.2% (77.5%) which means the disturbance would be more distributed across the entire matrix.

For USA and China, the shock is mainly concentrated around the originating country, making these two countries similar to Russia and Japan in that regard. However, some parts of the network for USA and China (i.e. the countries denoted with lighter colors in the column-

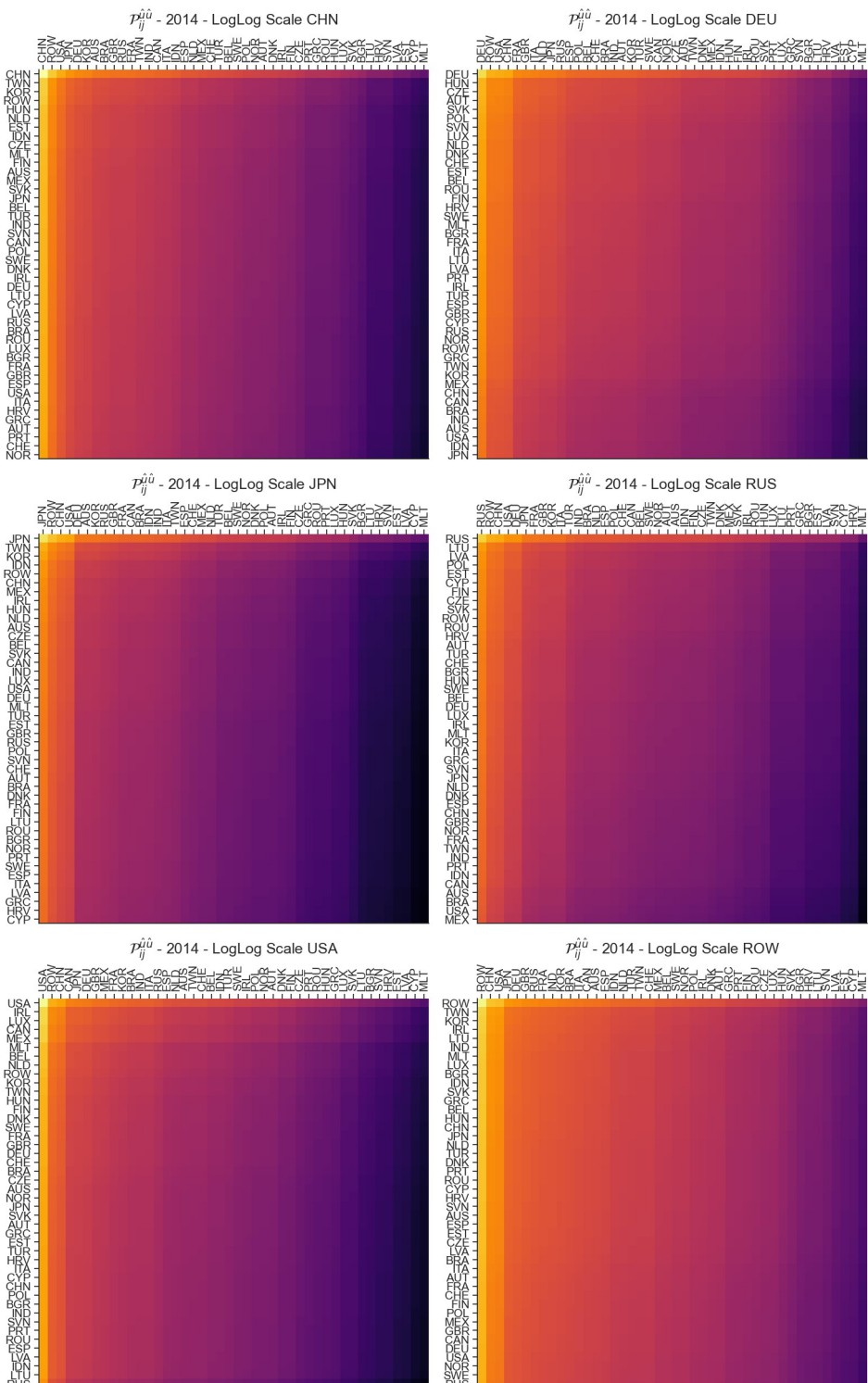

**Fig 4. Heatmap visualization of $\mathcal{P}_{ij}^{\hat{u}\hat{v}}$ obtained with $\hat{u} = \hat{v}$ for China (A), Germany (B), Japan (C), Russia (D), USA (E) and ROW (F) for 2014.** The visualization presents how the shock originating within the mentioned countries propagates across the world economy.

**Table 1. Distribution of sensitivity shocks originating in within a country.** We quantify how big of a percentage of the shock propagates within the country, in the top 10 trading partners and in the top half of the countries.

| Country | Within % | Input % | Output % | Top 10% | Top 22% |
|---|---|---|---|---|---|
| CHN | 56.1 | 3.4 | 38.2 | 76.4 | 87.6 |
| DEU | 32.6 | 9.4 | 45.0 | 70.0 | 88.2 |
| RUS | 48.9 | 19.5 | 22.6 | 85.0 | 94.1 |
| JPN | 62.3 | 17.5 | 15.8 | 91.8 | 96.2 |
| USA | 48.3 | 1.5 | 48.8 | 81.0 | 90.7 |
| ROW | 26.4 | 3.4 | 62.1 | 55.2 | 77.5 |

wise patterns) seem to be moderately affected by the shock propagation even though the network-wide impact is not as emphasized as in the case of Germany and ROW; similarly, other parts of the network (i.e. countries with darker colors) are rather isolated from the shock even though the isolation is not as strong as in the case of Russia and Japan.

$$\mathcal{P}_{ij}^{\hat{u}\hat{v}} = [p_{ij}^{\hat{u}\hat{v}}] \tag{18}$$

## Disturbance start input $\mathcal{P}_{::}^{u:}$ & output $\mathcal{P}_{::}^{:v}$ sensitivity

We can further aggregate the sensitivities of each link in $\mathcal{P}_{::}^{uv}$ to obtain higher level metrics. Two options for aggregation are available in this regard. Through the aggregation along the $V$ axis, we can get the average input sensitivity $\mathcal{P}_{::}^{u:}$ (as an exporter) of each economy $u$. On the other hand, by aggregating on the column along the $U$ axis, we can get the average output sensitivity $\mathcal{P}_{::}^{:v}$ (as an importer) of economy $v$. The aggregations $\mathcal{P}_{::}^{u:}$ and $\mathcal{P}_{::}^{:v}$ can be calculated as follows:

$$\mathcal{P}_{::}^{u:} = \left[ \sqrt[J^3]{\prod_v \prod_{i,j} p_{ij}^{uv}} \right]$$
$$\mathcal{P}_{::}^{:v} = \left[ \sqrt[J^3]{\prod_u \prod_{i,j} p_{ij}^{uv}} \right] \tag{19}$$

This results in two indices for each economy for each year, as shown in Figs 5–7. Out of the analyzed countries in the dataset, we can see that ROW, China, Germany, USA and Russia exhibit the largest overall impact in terms of input sensitivity. Similarly, USA, China and ROW exhibit dominant impact in terms of output sensitivity as can be noticed on Fig 2 as well. Furthermore, on Fig 5, we can see the dominance of USA's output sensitivity in 2000 which gradually decreases over the subsequent years until 2014. On the other hand, China's output sensitivity steadily increased from being near the average in 2000 (when it coincided with Germany's sensitivity) to even overtaking the USA in 2014 (Fig 6). Additionally, we can see a decrease in input sensitivity from 2008 to 2009 across all countries, likely due to outburst of the global financial crisis in 2008. Although we are focusing on 6 countries of interest within the paper, on Fig 7 we can see how the input and output sensitivity rankings for all countries in the WIOD dataset change through time.

## Disturbance end input $\mathcal{P}_{i:}^{::}$ & output $\mathcal{P}_{:j}^{::}$ sensitivities

Similarly, we can aggregate the sensitivities of each link across the $U$ and $V$ to $\mathcal{P}_{ij}^{::}$ to obtain higher level metrics. By taking the geometric mean of the row $J$, we can get the average input

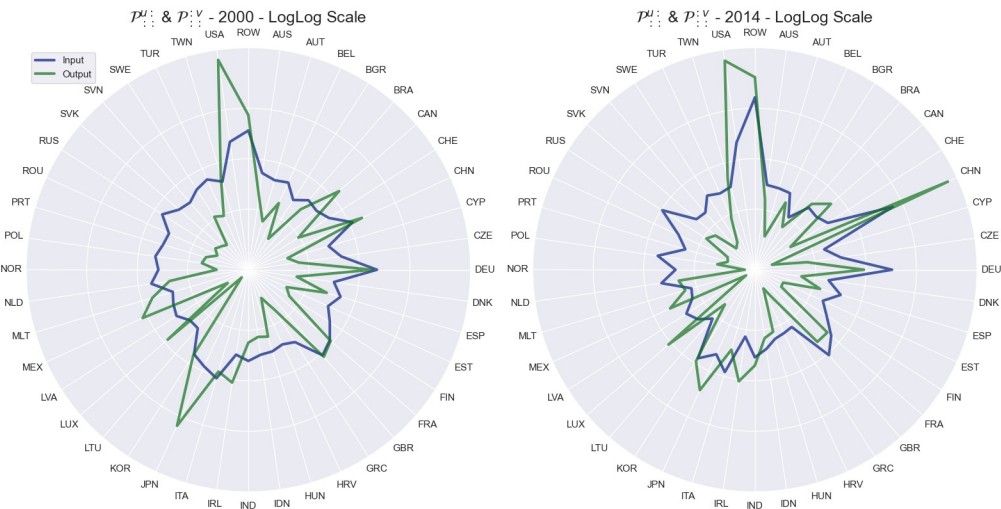

**Fig 5. Shock tensor aggregation $\mathcal{P}_{::}^{u:}$ (blue) and $\mathcal{P}_{::}^{:v}$ (green) for the years 2000 (A) and 2014 (B).** The visualization shows us the average impact of the shock which originates in a country as an exporter (blue) and importer (green).

volatility $\mathcal{P}_{i:}^{::}$ (as an exporter) of each economy. Similarly, by aggregating on the column *I*, we can get the average output volatility $\mathcal{P}_{:j}^{::}$ (as an importer) of a selected economy. The aggregations $\mathcal{P}_{i:}^{::}$ and $\mathcal{P}_{:j}^{::}$ can be calculated as follows:

$$\mathcal{P}_{i:}^{::} = \left[ \sqrt[J^3]{\prod_j \prod_{u,v} p_{ij}^{uv}} \right]$$

$$\mathcal{P}_{:j}^{::} = \left[ \sqrt[J^3]{\prod_i \prod_{u,v} p_{ij}^{uv}} \right]$$

(20)

This results in two indices for each economy for each year, as shown in Figs 8–10. The input volatility is noticeably more balanced across the countries compared to output volatility,

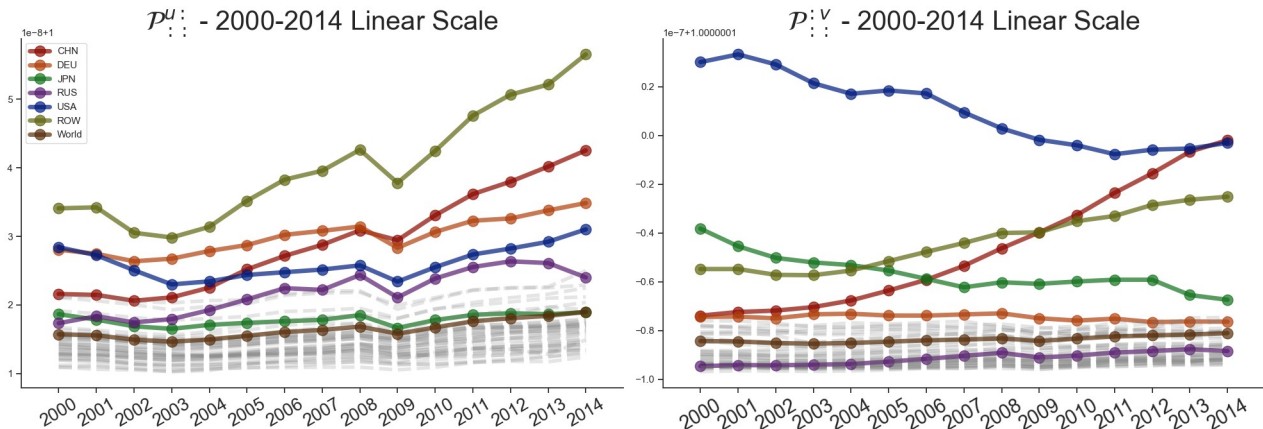

**Fig 6. Shock tensor aggregation $\mathcal{P}_{::}^{u:}$ and $\mathcal{P}_{::}^{:v}$ for the countries China, Germany, Japan, Russia and USA as well as ROW over the period between 2000 and 2014.** The world average is represented in brown, with every other country shown in gray. The visualization shows us the time series of the average sensitivity where the originating country of the shock is the exporter (A) and importer (B).

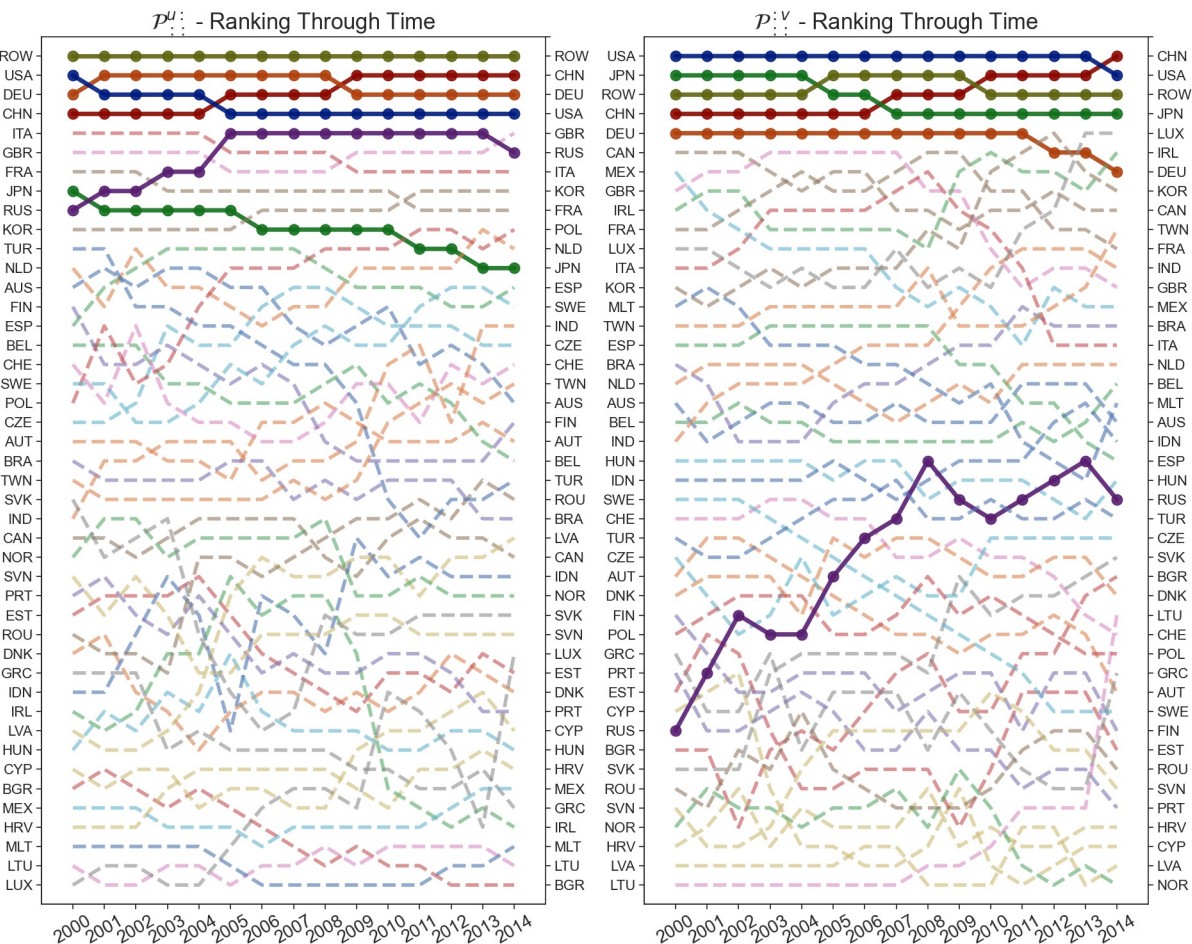

**Fig 7. Rankings of the shock tensor aggregation $\mathcal{P}^{u:}_{::}$ and $\mathcal{P}^{:v}_{::}$ over the period between 2000 and 2014.** The visualization shows us the how the rankings of the disturbance start sensitivity change through time.

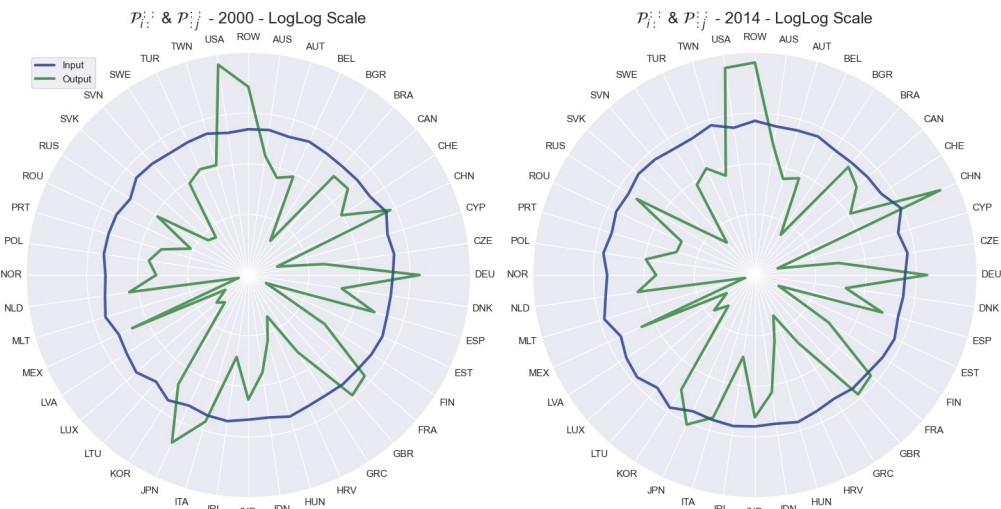

**Fig 8. Shock tensor aggregation $\mathcal{P}^{::}_{i:}$ (blue) and $\mathcal{P}^{::}_{:j}$ (green) for the years 2000 (A) and 2014 (B).** The visualization shows us the average impact of the shock which finished in a country as an exporter (blue) and importer (green).

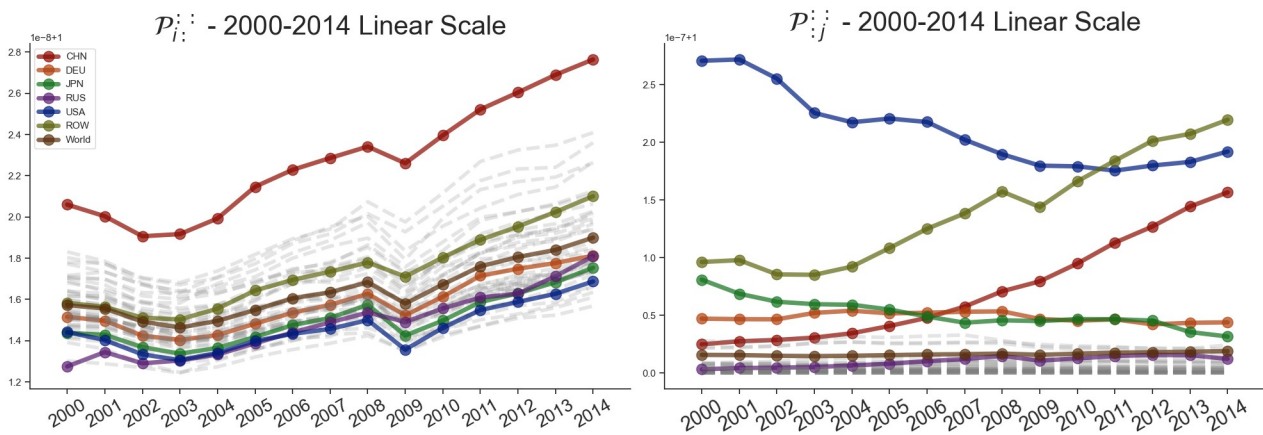

**Fig 9. Shock tensor aggregation $\mathcal{P}_{i:}^{::}$ and $\mathcal{P}_{:j}^{::}$ for the countries China, Germany, Japan, Russia and USA as well as ROW over the period between 2000 and 2014.** The world average is represented in brown, with every other country shown in gray. The visualization shows us the time series of the average sensitivity where the consuming country of the shock is the exporter (A) and importer (B).

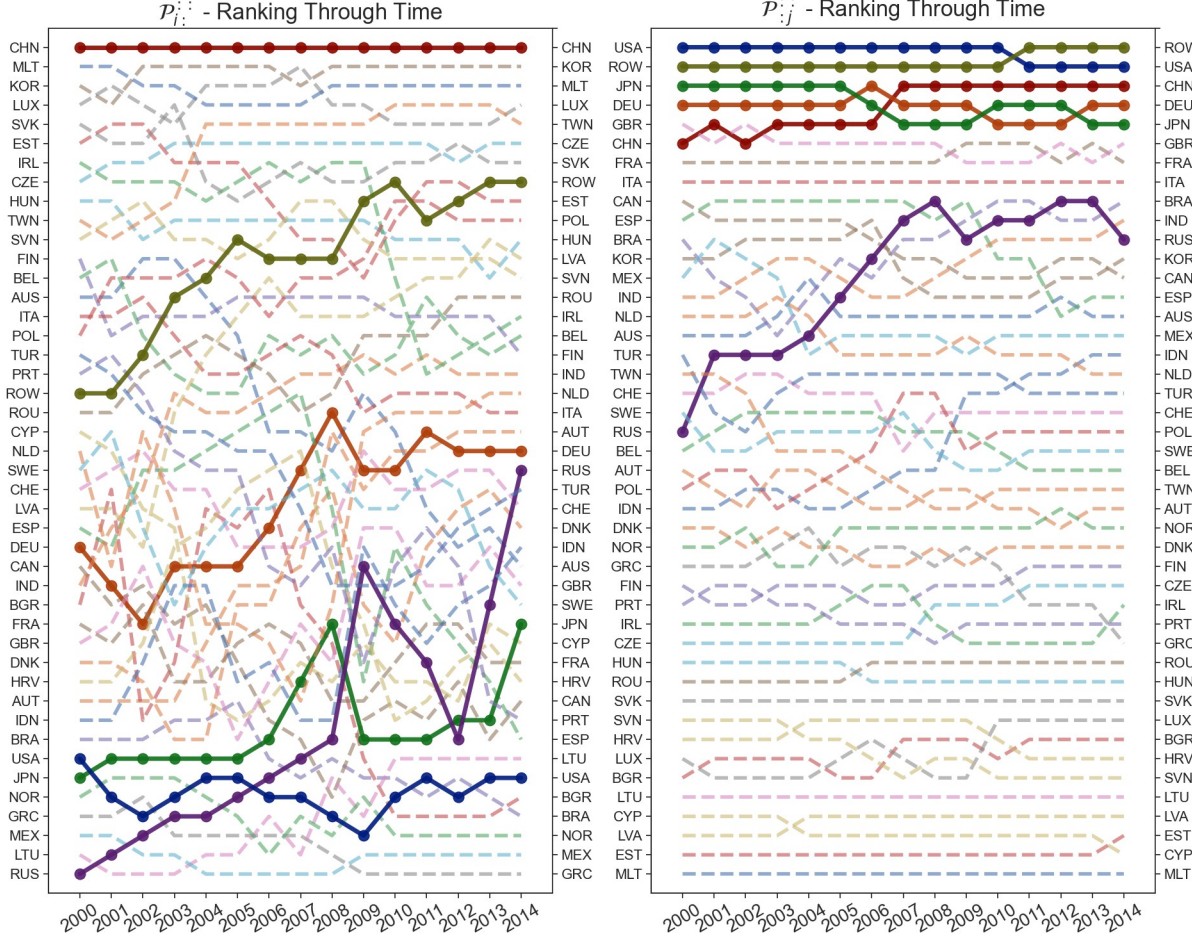

**Fig 10. Rankings of the shock tensor aggregation $\mathcal{P}_{i:}^{::}$ and $\mathcal{P}_{:j}^{::}$ over the period between 2000 and 2014.** The visualization shows us the how the rankings of the disturbance end sensitivity change through time.

however, for most countries, the average volatility as an exporter is much higher than the volatility as an importer (Fig 8). Interestingly, here China has the largest input volatility throughout the entire time period. The USA starts with the largest output sensitivity by far, however, it has a significant drop over the fourteen-year period and is overtaken by the ROW and China economies which unlike most countries achieve a significant growth (Fig 9). Finally, on Fig 10 we can see the input and output sensitivity rankings for all countries in the WIOD dataset change through time. While the input volatility show a high instability through time, this is most likely due to the minute difference within the input sensitivities discussed about previously. On the other hand, output volatility show the least changes through time, with a few exceptions such as the rise in the output volatility ranking of Russia.

## Disturbance end fixed input $\mathcal{P}_{i:}^{\hat{u}\hat{v}}$ & output $\mathcal{P}_{:j}^{\hat{u}\hat{v}}$ sensitivity

The matrix $\mathcal{P}_{ij}^{\hat{u}\hat{v}}$ can be further aggregated to the input and output level, resulting in the operators $\mathcal{P}_{i:}^{\hat{u}\hat{v}}$ and $\mathcal{P}_{:j}^{\hat{u}\hat{v}}$ respectively. These two aggregations can be calculated as follows:

$$\mathcal{P}_{i:}^{\hat{u}\hat{v}} = \left[ \sqrt[J]{\prod_j p_{ij}^{\hat{u}\hat{v}}} \right]$$
$$\mathcal{P}_{:j}^{\hat{u}\hat{v}} = \left[ \sqrt[J]{\prod_i p_{ij}^{\hat{u}\hat{v}}} \right]$$

(21)

The results are shown on Fig 11 for the years of 2000 and 2014.

## Overall sensitivity $\mathcal{P}_{::}^{::}$

The aggregation to the highest level of detail is the one mentioned above where a geometric mean is taken along all axes of the tensor. This aggregation can be calculated as follows:

$$\mathcal{P}_{::}^{::} = \left[ \sqrt[J^4]{\prod_{i,v}\prod_{u,v} p_{ij}^{uv}} \right]$$

(22)

resulting in an index for each year which as shown on Fig 12. We similarly calculate the aggregation of the tensor $\mathcal{Z}$ where we take the arithmetic mean along all axes resulting in an overall index for input flows over time $\mathcal{Z}_{::}^{::}$. While the graphs of the two functions are overall similar, including the considerable downturn in 2008, there are also differences that can be noticed. For example, the first difference is hat $\mathcal{Z}_{::}^{::}$ is higher than $\mathcal{P}_{::}^{::}$ at the beginning of the century but decreases in 2000–2003 whereas $\mathcal{P}_{::}^{::}$ increases slowly on the same period. $\mathcal{Z}_{::}^{::}$ becomes higher than $\mathcal{P}_{::}^{::}$ in 2003 when $\mathcal{P}_{::}^{::}$ reaches its lowest value in the analyzed period 2000–2014. The second difference can be evidenced towards the end of the time period. While both functions are increasing throughout 2000–2014, it can be seen that they have different slopes towards the end of the period; $\mathcal{Z}_{::}^{::}$ grows slower than $\mathcal{P}_{::}^{::}$ in 2011–2014, and their values ultimately coincide in 2014. Over the periods 2003–2007 and 2009–2011 the slopes of the functions are comparable.

## Conclusion

We study shock distributions in the world economy by quantifying the changes in network linkages using sensitivity analysis. We model the world economy as an interconnected network according to the recent 2016 release of the World Input-Output Database (WIOD). Following this model, we define a shock tensor with the aim of evaluating the impact of link sensitivity

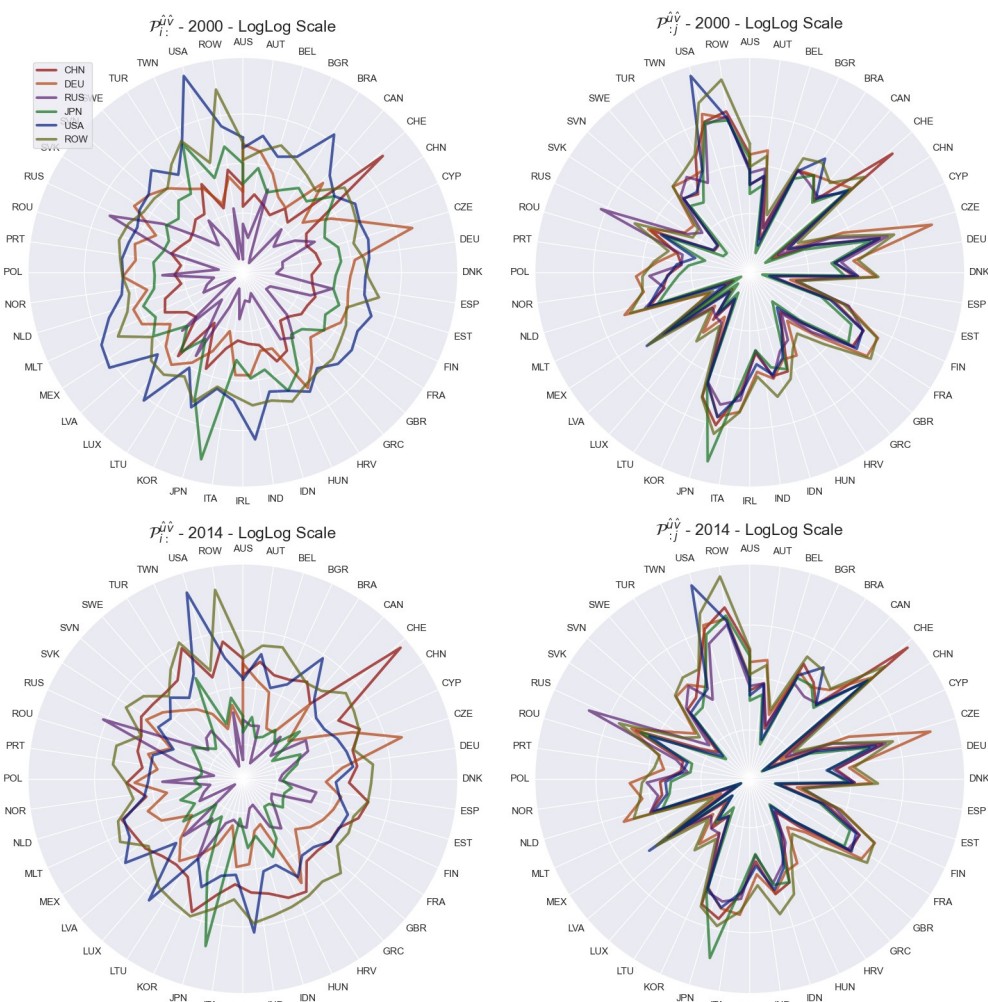

**Fig 11. Shock tensor aggregation for $\mathcal{P}_{i:}^{\hat{u}\hat{v}}$ (top row) and $\mathcal{P}_{:j}^{\hat{u}\hat{v}}$ (bottom row) for the years of 2000 (left column) and 2014 (right column) for China, Germany, Russia, Japan, USA and ROW.** The visualization presents how the shock originating within the mentioned countries as propagates across the world economy.

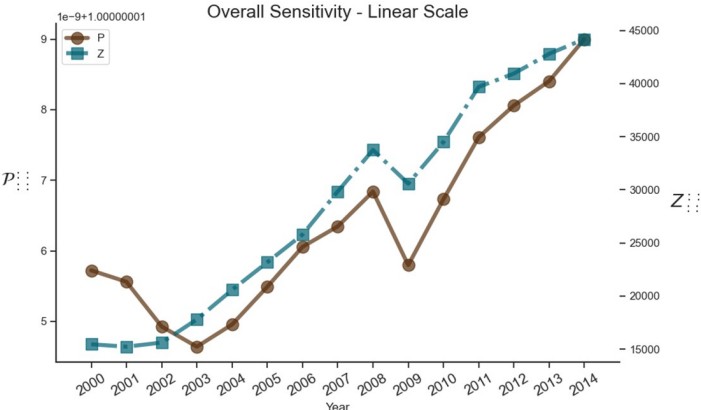

**Fig 12. Shock tensor aggregation $\mathcal{P}_{::}$ (brown), which shows the overall sensitivity of the world-input network, and $\mathcal{Z}_{::}$ (teal), which shows the overall trade of the world-input network, in the period from 2000 to 2014.**

on the shock intensity and propagation through the network. We show that the shock tensor can be used to aggregate trade data at various levels of granularity. This leads to illustrative visualization results that reveal patterns of shock distributions and relationships among different countries. The results could be helpful to policy makers when analyzing trade relationships between countries, assessing risks and making informed decisions.

## Supporting information

**S1 Appendix. Theoretical model.**
(PDF)

**S2 Appendix. Proof of the theorems.**
(PDF)

## Author Contributions

**Conceptualization:** Ljupco Kocarev.

**Data curation:** Viktor Domazetoski, Maryan Rizinski.

**Formal analysis:** Viktor Domazetoski, Ljupco Kocarev.

**Funding acquisition:** Dimitar Trajanov, Ljupco Kocarev.

**Methodology:** Ljupco Kocarev.

**Project administration:** Dimitar Trajanov, Ljupco Kocarev.

**Software:** Viktor Domazetoski, Maryan Rizinski.

**Supervision:** Dimitar Trajanov, Ljupco Kocarev.

**Visualization:** Viktor Domazetoski.

**Writing – original draft:** Ljupco Kocarev.

**Writing – review & editing:** Viktor Domazetoski, Maryan Rizinski, Dimitar Trajanov, Ljupco Kocarev.

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
