## [Decision Letter · Decision Letter 0]

27 Mar 2023

PONE-D-22-33523Sensitivity analysis of shock distributions of network linkages in the world economyPLOS ONE

Dear Dr. Domazetoski,

Thank you for submitting your manuscript to PLOS ONE. After careful consideration, we feel that it has merit but does not fully meet PLOS ONE’s publication criteria as it currently stands. Therefore, we invite you to submit a revised version of the manuscript that addresses the points raised during the review process.

We look forward to receiving your revised manuscript.

Kind regards,

Emilia Lamonaca

Academic Editor

PLOS ONE

Journal Requirements:

Additional Editor Comments:

The conclusions (and the comment to the results) fail to show how the paper speaks with the previous literature. The reader expects to see precise comparisons of the findings derived from this study with those already established in the literature.

Reviewers' comments:

Reviewer's Responses to Questions

**Comments to the Author**

1. Is the manuscript technically sound, and do the data support the conclusions?

Reviewer #1: Partly

2. Has the statistical analysis been performed appropriately and rigorously? 

Reviewer #1: Yes

3. Have the authors made all data underlying the findings in their manuscript fully available?

Reviewer #1: No

4. Is the manuscript presented in an intelligible fashion and written in standard English?

Reviewer #1: Yes

5. Review Comments to the Author

Reviewer #1: GENERAL

In the context of a globalized and highly interconnected world economy, the authors explore the impact of shocks in global value chain networks on countries’ output supply and input demand. They aim to provide a new tool to assess the distribution of these impacts between countries. They model countries’ interconnectedness through a discrete-time absorbing Markov chain – for which the model is developed in a previous paper (Kostoka et al., 2020). The contribution of the present paper is to add a shock tensor to this model. This allows the authors to derive patterns of global shock distribution, which are quantified through the use of the University of Groeningen’s World Input-Output Database (WIOD). While the data is not submitted directly in the manuscript, it is publicly available for download.

The authors present an interesting analysis of the distribution of these shocks. The model is presented in a clear and concise manner and seems appropriate for GVC analysis. However, it is not clear what the authors want to highlight as their contribution in this paper. There appear to be two options here: either the contribution is methodological, or it is empirical.

If the contribution is purely methodological, then there needs to be more work done to show the advantages of their model compared to existing literature on this topic and other approaches – including compared to their own previous work (Kostoka et al., 2020). The results should then be oriented to highlight the advantages of this methodology.

If the contribution is empirical, then the results need to be interpreted a lot more and policy implications need to be drawn. In this case, it would be very helpful to use a case study of a particular country or to explore one particular industry and to answer one or two topical questions. E.g.: What kinds of repercussions would a shock in China’s textile industry have on GVCs ?

Aside from this, there are several major comments that need to be addressed before the paper can be published. These are detailed below. Minor comments are also suggested, along with a list of small language/typing corrections.

MAJOR COMMENTS

1. The contextualization for the paper that is given in the introduction needs to be strengthened. Some elements that might make your argument stronger:

- Your literature review needs to be developed more. Currently, you just list the papers that are in the literature on shock propagation, without really going into the contributions of different authors/strands of the literature. This makes it hard to understand exactly where your paper is positioned with regards to the rest of the literature on GVCs.

- Additionally, you say (p.2, l.44-47) that “there is yet work to be done in the literature to examine … the extent to which shocks originating from a country or industry affect other countries or industries”. Said in this way, it seems to the reader that there is no literature studying the impact of shocks on GVCs. However, there is a large literature on this topic – although the methodologies that are used may differ from the one used in this paper. See for instance Wenz & Willner (2022), Climate impacts and global supply chains: an overview, a chapter in a handbook which discusses the literature on these shocks in the context of climate impacts; Qin et al. (2020) Covid-19 Shock and Global Value Chains: Is there a substitution for China? and Gershel et al. (2020) Propagation of shocks in the global value chains: the coronavirus case. that study GVC shock propagation in the case of the COVID crisis, … These are just examples, but it would be necessary to give an overview of what has been done before to study GVC shock propagation.

- When you state “Most macroeconomic models typically derive from the Leontief’s classical work on input-output tables that characterize global production networks”, you should cite some of the most important papers that have actually done this, or a literature review on this topic to support your statement.

- The “diversification argument” is just mentioned but not explained at all. However, it seems that it is an important concept to justify your research, since you state in the paragraph right after its mention that you aim to tackle the limitations of this argument (namely linkages between industries as propagation channels). Given its apparent importance, you should define this argument, provide some background literature on it and explain its limitations in more detail (+ maybe cite other authors that have worked on these limitations).

2. The policy relevance of your results is also not clear. You simply write “The results show trends about the aggregate effects of shock distributions which could be helpful to policy makers in assessing risks arising from country or industry interdependence and trade relationships” but this is quite vague and does not explain how your methodology specifically provides insights that would be useful – especially in contrast with other types of studies.

3. Your results are presented in a way that makes it unclear what exactly you want to highlight. There is almost no interpretation of the results you present, or policy implications that are derived. For example, you say for figure 2 that “The highest values in 2000 can be noticed in the links USA-USA and ROW-USA, while the highest values in 2014 can be noticed in the links China- China and ROW-China.” What does this imply for these countries? What are the risks? What kinds of policies should policymakers be thinking about applying as a response?

The same comment goes for all the figures that are presented in the results section – while they are sometimes described, they are not interpreted. This would go a long way to help the reader understand the importance of your results. Additionally, it might be interesting to take a particular case / example to illustrate how your results can be interpreted. For instance, take one of the countries you are studying and identify which of its sectors are the most sensitive to shocks from which countries – then derive policy implications for policymakers in this country.

+ the inter-country heterogeneity in your results is interesting– can you interpret it more? What does it say about the vulnerability of different countries?

+ the fact that there are differences in the impact of a shock for a country if it is an importer in the GVC and if it is an exporter is also an interesting result that is not commented at all. It is especially visible in figures 5 and 8.

4. In your description of figure 4, you state that “for Germany and ROW the shock is propagated more considerably throughout the network when compared with Russia and Japan where the shock is concentrated around the originating country”. However, this is really not clear in the figures your present. While the ROW figure does seem lighter than the others, the figure for Germany is not that much lighter than Russia for instance. If you really want to make that comparison, would it be possible to add the interpretation of quantitative results? Rather than purely basing your analysis on a visual interpretation of the color scheme, where the differences are not very pronounced.

5. There is a problem in figure 5. In your description of the figure, you state that “China’s output sensitivity steadily increased from being near the average in 2000 … to even overtaking the USA in 2014” � looking at the right-hand-side of figure 5, this is not what is shown. Indeed, your graph shows that Switzerland overtakes the USA in 2014, not China. This looks like it might just be a discrepancy in the axes and the labels of the graphs.

6. Figure 7 is not commented at all.

7. In the appendix, you could add more details to the steps described to derive your model.

8. Your references in the appendix are not correctly formatted (there are “?” in lieu of all references).

MINOR COMMENTS

1. The introduction begins with and is substantially (about a third of it) devoted to a discussion on the links between competition, globalization, and international trade dynamics. However, the competition aspects of globalization are not really addressed anywhere in the rest of the paper. It might be better to refocus the introduction on the risks of globalization – i.e., the heart of the model & results. Giving example of these risks would also be beneficial (the COVID crisis is a very obvious one).

2. You can shorten your description of the WIOD by only retaining the main elements that are useful for your model/analysis. Interested readers can refer back to the database’s documentation to get more information if needed.

3. You don’t describe the second database you use in your “Data” section – the Total Factor Productivity data from the Conference Board Total Economy Database.

4. You could explain why you work specifically within the country world-input network, rather than the other 2 possible variations. You choose this variation in particular without really detailing why it is more relevant than the others.

5. A(1) is not defined in your main paper - you only define A(2) and A(3) explicitly.

6. You don’t explain why it is “more reasonable” to use sensitivity as an indicator rather than elasticity in the case of network linkages. Is this something that is standard in the literature? What are the advantages?

7. You don’t provide any preview of your qualitative results in the introduction (or in the abstract). You should add highlights of the elements that are most significant and how they relate to existing literature (are they surprising? standard?).

8. In your comments on figure 12, you note that there is a “considerable downturn in 2008” but don’t try to explain this. It seems rather counterintuitive as one could think that during the crisis, countries were even more dependent on GVCs, and therefore more sensitive to shocks from other countries. Additionally, you describe the differences between the Z and P tensors, but do not interpret these differences. What do they tell us about overall risk sensitivity in GVC networks?

LANGUAGE / MISSPELLINGS

1. P.1, l.4: “pervasive” has a strong negative connotation that does not seem appropriate here

2. P.2, l.29: “growingly integrated” is clumsy, you might want to change to “increasingly integrated”

3. P.2 l.29-32: “GVCs have been used … for International Development.” This sentence needs to be rephrased.

4. P. 2, l.37: “draws attention that” words missing

5. P.2, l. 52: “lenghts" is misspelled

6. P.3, l. 106: you use the WIOT abbreviation without having ever defined it (should be at the beginning of that paragraph when you spell out “World Input-Output Tables”

7. P.4, l.111: “use” typed twice

8. P.6, l.174: you used the wrong epsilon symbol

9. P.7, l.203: “the specified axis such the calculation…”: missing “as”

10. P.7, l.214 + p.8, l.223 : “an J x J matrix” - should be “a J x J matrix”

6. PLOS authors have the option to publish the peer review history of their article (what does this mean?). If published, this will include your full peer review and any attached files.

Reviewer #1: No

---

## [Author Response · Author response to Decision Letter 0]

2 Oct 2023

Dear editor,

We would like to thank the Reviewer for the very useful comments. We believe that these comments will contribute to the improvement of the overall presentation of our work. We have carefully addressed all the risen points and amended the manuscript accordingly, and believe that the revised version of our manuscript satisfies all criteria for publication in your journal. Please find the reply to the comments below. In the revised version of the paper, deletions are marked in red, and insertions are marked in blue.

Viktor Domazetoski,

on behalf of the authors

Reviewer #1:

1. In the context of a globalized and highly interconnected world economy, the authors explore the impact of shocks in global value chain networks on countries’ output supply and input demand. They aim to provide a new tool to assess the distribution of these impacts between countries. They model countries’ interconnectedness through a discrete-time absorbing Markov chain – for which the model is developed in a previous paper (Kostoka et al., 2020). The contribution of the present paper is to add a shock tensor to this model. This allows the authors to derive patterns of global shock distribution, which are quantified through the use of the University of Groeningen’s World Input-Output Database (WIOD). While the data is not submitted directly in the manuscript, it is publicly available for download.

The authors present an interesting analysis of the distribution of these shocks. The model is presented in a clear and concise manner and seems appropriate for GVC analysis. However, it is not clear what the authors want to highlight as their contribution in this paper. There appear to be two options here: either the contribution is methodological, or it is empirical.

If the contribution is purely methodological, then there needs to be more work done to show the advantages of their model compared to existing literature on this topic and other approaches – including compared to their own previous work (Kostoka et al., 2020). The results should then be oriented to highlight the advantages of this methodology.

If the contribution is empirical, then the results need to be interpreted a lot more and policy implications need to be drawn. In this case, it would be very helpful to use a case study of a particular country or to explore one particular industry and to answer one or two topical questions. E.g.: What kinds of repercussions would a shock in China’s textile industry have on GVCs ?

Aside from this, there are several major comments that need to be addressed before the paper can be published. These are detailed below. Minor comments are also suggested, along with a list of small language/typing corrections.

Answer: The main contribution of the paper is to propose a novel methodology for evaluating shock distributions across the global economy as modeled by the WIOD dataset. The methodology consists of two new theorems. The experimental results are included to show that various insights can be obtained from applying the methodology on the WIOD dataset. To help position our model within the broader literature we added 5 references and the following paragraph in the introduction of the manuscript:

“Sensitivity analysis of a country Input-Output (IO) models has a long history and has been proceeded along three paths. The first path considers the extent to which errors interact through matrix inversion to bias the results and here we mention some results in this direction: Simonovitz (1975) discusses underestimation and overestimation of the Leontief Inverse, while Lahiri and Satchel (1985) derive necessary and sufficient conditions for the over- and underestimation of inverse elements, assuming that prices are the source of the stochastic errors. The second path builds on the results of Sherman and Morrison (1950) who traced the effects of a discrete change in a single technical coefficient. The third path explores the feasibility of using Monte Carlo simulation to study stochastic error propagation through IO models, which has been pioneered by Clark W. Bullard and Anthony V. Sebald in a series of papers, see, for example (Bullard and Sebald, 1977) and (Bullard and Sebald, 1978).”

Simonovits, A., A Note on the Underestimation and Overestimation of the Leontief Inverse, Econometrica 43 (1975) 493-498.

Lahiri, Sajal and Steve Satchell, Underestimation and Overestimation of the Leontief Inverse Revisited, Economics Letters 18 (1985), 181-186.

Sherman, Jack, and Winifred J. Morrison, Adjustment of an Inverse Matrix Corresponding to a Change in One Element of a Given Matrix, Annals of Mathematical Statistics 21 (1950), 124-127.

Bullard, Clark W., and Anthony V. Sebald, Effects of Parametric Uncertainty and Technological Change on Input- Output Models, The Review of Economics and Statistics (1) (1977), 75-81.

Bullard, Clark W., and Anthony V. Sebald, “Monte Carlo Sensitivity Analysis of Input-Output Models”, The Review of Economics and Statistics, Vol. 70, No. 4 (Nov., 1988), pp. 708-712.

While previous work in Kostoska et al. worked with global value chains and used the same data, it utilized discrete-time absorbing Markov chains to measure the positioning of countries and industries in GVC, and didn’t include any sensitivity analysis. To make this information clearer, we expanded on our text in the introduction by adding more information on the Kostoska et al. paper: 

“This paper builds on the prior research in [7] where discrete-time absorbing Markov chains are proposed to model the structure and interdependence among country-industry pairs of the world economy. Several novel properties are designed based on the discrete-time absorbing Markov chains approach with the aim of evaluating the volatility and risk when shaping production chain lengths. In addition, the study shows that the input and output chains exhibit exactly the same quasi-stationary product distribution, meaning that the time spent in a state before absorption is invariant to the changes of the network type [7]. The paper also suggests several global metrics, including the probability distribution of global value added/final output, provide guidance for policy makers when estimating the resilience of world trading system and forecasting the macroeconomic developments.”

2. The contextualization for the paper that is given in the introduction needs to be strengthened. Some elements that might make your argument stronger:

- Your literature review needs to be developed more. Currently, you just list the papers that are in the literature on shock propagation, without really going into the contributions of different authors/strands of the literature. This makes it hard to understand exactly where your paper is positioned with regards to the rest of the literature on GVCs.

Answer: “To help position our model within the broader literature we added the following paragraph in the introduction of the manuscript mentioned also in the first answer:

“Sensitivity analysis of a country Input-Output (IO) models has a long history and has been proceeded along three paths. The first path considers the extent to which errors interact through matrix inversion to bias the results and here we mention some results in this direction: Simonovitz (1975) discusses underestimation and overestimation of the Leontief Inverse, while Lahiri and Satchel (1985) derive necessary and sufficient conditions for the over- and underestimation of inverse elements, assuming that prices are the source of the stochastic errors. The second path builds on the results of Sherman and Morrison (1950) who traced the effects of a discrete change in a single technical coefficient. The third path explores the feasibility of using Monte Carlo simulation to study stochastic error propagation through IO models, which has been pioneered by Clark W. Bullard and Anthony V. Sebald in a series of papers, see, for example (Bullard and Sebald, 1977) and (Bullard and Sebald, 1978).”

3. Additionally, you say (p.2, l.44-47) that “there is yet work to be done in the literature to examine … the extent to which shocks originating from a country or industry affect other countries or industries”. Said in this way, it seems to the reader that there is no literature studying the impact of shocks on GVCs. However, there is a large literature on this topic – although the methodologies that are used may differ from the one used in this paper. See for instance Wenz & Willner (2022), Climate impacts and global supply chains: an overview, a chapter in a handbook which discusses the literature on these shocks in the context of climate impacts; Qin et al. (2020) Covid-19 Shock and Global Value Chains: Is there a substitution for China? and Gershel et al. (2020) Propagation of shocks in the global value chains: the coronavirus case. that study GVC shock propagation in the case of the COVID crisis, … These are just examples, but it would be necessary to give an overview of what has been done before to study GVC shock propagation.

Answer: “We apologize on the oversight, we agree with the reviewer that the review of shocks on GVCs was inadequate. To show previous work on how the impact of shocks on GVCs has been studied, we added 7 references the following paragraph in the introduction:

“Sensitivity analysis of World-Input-Output models and/or multi-regional input–output (MRIO)

models has also recently attracted great deal of interest due to the emergence of GVCs. To assess the impact of shocks in a world of global value chains, several models and approaches have been developed. Here we mention several, thus, for example, (Caliendo and Parro, 2014) build a Ricardian model with sectoral linkages and trade in intermediate goods to quantify the welfare effects from tariff changes, while (Anderson et al., 2015) offers a structural gravity approach to quantify output and welfare effects. The paper (Vandenbussche, Connell and Simons, 2020) uses the sectoral World

Input–Output Database (WIOD) to evaluate the impact in terms of value added and employment of different scenarios of Brexit for 56 industries in the 27 Member States of the European Union, as well as the United Kingdom. Gerschel, Martinez & Mejean, (2020) employ WIOD dataset to measure the share of a particular sector in a particular country as a source in the gross output of a particular sector of a given country. Using the Leontief inverse matrix, the authors measure how the gross output of each sector from each country is exposed to shocks affecting China, both directly and indirectly. Using the World Input-Output Database (WIOD), Mandel and Veetil (2020b) study the effects of national lockdowns on global GDP in a non-equilibrium framework. Pichler and Farmer (2022) combine WIOD framework and the approach of del Rio-Chanona et al. (2020) to compute supply shocks for every industry during a lockdown of Covid-19 in Germany, Italy, and Spain.”

Caliendo, L., & Parro, F. (2014). Estimates of the trade and welfare effects of NAFTA. The Review of Economic Studies, 82(1), 1–44. https://doi.org/10.1093/restu d/rdu035

Anderson, J. E., Larch, M., & Yotov, Y. V. (2015). Growth and trade with frictions: A structural estimation framework. Technical report. National Bureau of Economic Research.

Vandenbussche, H., Connell, W., & Simons, W. (2022). Global value chains, trade shocks and jobs: An application to Brexit. The World Economy, 45(8), 2338-2369.

Gerschel, E., Martinez, A., & Mejean, I. (2020). Propagation of shocks in global value chains: the coronavirus case. Notes IPP, (53).

Pichler, A., & Farmer, J. D. (2022). Simultaneous supply and demand constraints in input–output networks: the case of Covid-19 in Germany, Italy, and Spain. Economic Systems Research, 34(3), 273-293.

Mandel A., & Veetil V. P. (2020b). The economic cost of Covid lockdowns: An out-of-equilibrium analysis. Economics of Disasters and Climate Change, 4(3), 431–451. https://doi.org/10.1007/s41 885-020-00066-z

del Rio-Chanona R. M., Mealy P., Pichler A., Lafond F., & Farmer J. D. (2020). Supply and demand shocks in the Covid-19 pandemic: An industry and occupation perspective. Oxford Review of Economic Policy, 36(1), 94–137. https://doi.org/10.1093/oxrep/graa033

4. When you state “Most macroeconomic models typically derive from the Leontief’s classical work on input-output tables that characterize global production networks”, you should cite some of the most important papers that have actually done this, or a literature review on this topic to support your statement.

Answer: To support our statement, we refer to the added paragraph in the introduction section mentioned in the first our answer.

5. The “diversification argument” is just mentioned but not explained at all. However, it seems that it is an important concept to justify your research, since you state in the paragraph right after its mention that you aim to tackle the limitations of this argument (namely linkages between industries as propagation channels). Given its apparent importance, you should define this argument, provide some background literature on it and explain its limitations in more detail (+ maybe cite other authors that have worked on these limitations).

Answer: As per your request, we have added the following additional information on the diversification argument in the introduction: 

“…The 'diversification argument' posits that within an economy comprised of n industries experiencing independent shocks, the magnitude of aggregate fluctuations would roughly scale inversely with the square root of n. This suggests that when examining highly detailed or disaggregated levels of the economy, individual shocks tend to have relatively minor effects on overall fluctuations. However, the diversification argument has several limiting assumptions such as the independence of shocks and industry homogeneity. Notably, it also takes no account of linkages between industries that can also serve as a channel for shock propagation through the network.”

6. The policy relevance of your results is also not clear. You simply write “The results show trends about the aggregate effects of shock distributions which could be helpful to policy makers in assessing risks arising from country or industry interdependence and trade relationships” but this is quite vague and does not explain how your methodology specifically provides insights that would be useful – especially in contrast with other types of studies.

Answer: As the paper’s main focus is primarily methodological and not experimental, we mostly stick to a methodological and descriptive interpretation of the results. However, we also release the source code and several illustrative examples so that interested readers can obtain results of their choice that would help them in policy decisions.

7. Your results are presented in a way that makes it unclear what exactly you want to highlight. There is almost no interpretation of the results you present, or policy implications that are derived. For example, you say for figure 2 that “The highest values in 2000 can be noticed in the links USA-USA and ROW-USA, while the highest values in 2014 can be noticed in the links China- China and ROW-China.” What does this imply for these countries? What are the risks? What kinds of policies should policymakers be thinking about applying as a response?

The same comment goes for all the figures that are presented in the results section – while they are sometimes described, they are not interpreted. This would go a long way to help the reader understand the importance of your results. Additionally, it might be interesting to take a particular case / example to illustrate how your results can be interpreted. For instance, take one of the countries you are studying and identify which of its sectors are the most sensitive to shocks from which countries – then derive policy implications for policymakers in this country.

+ the inter-country heterogeneity in your results is interesting– can you interpret it more? What does it say about the vulnerability of different countries?

+ the fact that there are differences in the impact of a shock for a country if it is an importer in the GVC and if it is an exporter is also an interesting result that is not commented at all. It is especially visible in figures 5 and 8.

Answer: Given the fundamental nature of this paper, which primarily centers around elucidating our methodology, we opted not to engage in an in-depth discussion of result interpretations or policy implications. Our primary concern was to maintain the paper's readability and accessibility for our readers. Expanding upon these aspects would have significantly extended the paper's length, potentially making it less approachable for its intended audience. However, it's important to note that while we refrained from delving into these areas in this particular manuscript, we recognize their significance. We therefore have plans to address these facets in future research endeavors, both at the level of different aggregations and of different countries of interest. 

8. In your description of figure 4, you state that “for Germany and ROW the shock is propagated more considerably throughout the network when compared with Russia and Japan where the shock is concentrated around the originating country”. However, this is really not clear in the figures your present. While the ROW figure does seem lighter than the others, the figure for Germany is not that much lighter than Russia for instance. If you really want to make that comparison, would it be possible to add the interpretation of quantitative results? Rather than purely basing your analysis on a visual interpretation of the color scheme, where the differences are not very pronounced.

Answer: We thank the reviewer for this comment. We agree that it would be beneficial to also include some quantitative results on top of the illustrative examples. Therefore, we added the following paragraph and table within the above-mentioned section: 

“… To quantify this, we can calculate the percentage of the shock contained within a subset of the matrix by normalizing log(pˆuˆv ij ) by the sum of the matrix. We do this calculation for the sensitivity within the country pˆuˆu uu), where the country acts as an exporter pˆuˆu

uj ) for j in 1, ..., n and j 6 = u, where the country acts as an importer pˆuˆu iu ) for i in 1, ..., n and i 6 = u. Additionally, we look at top 10 and 22 upper left values of the matrices shown in Fig. 4. The results are presented in Table 1. Now we can further see how 62.3% of the shock which originates within Japan ends in Japan, while this is at 26.4% and 32.6% for ROW and Germany. If we look at the input percentages, we see how much of the shock flows into countries which import from the fixed country, with Russia’s trade partners being the most affected. Similarly, the output percentages show what percentage of the shock is distributed to countries which act as exporters in the scenario. These values are much higher than the input sensitivities across all countries, with highest values for the Rest of World Model. The top 10 and 22 percentages show us the overall distribution of the shock. Again, Japan and Russia contain 91.8% (96.2%) and 85.0% (94.1%) for the top 10 (top 22) percentages, showing a less global impact on the economy, while for ROW these values are 55.2% (77.5%) which means the disturbance would be more distributed across the entire matrix.” 

9. There is a problem in figure 5. In your description of the figure, you state that “China’s output sensitivity steadily increased from being near the average in 2000 … to even overtaking the USA in 2014” ◊ looking at the right-hand-side of figure 5, this is not what is shown. Indeed, your graph shows that Switzerland overtakes the USA in 2014, not China. This looks like it might just be a discrepancy in the axes and the labels of the graphs.

Answer:

This was due to a bug in the visualization code, and it also occurs in the same kind of visualization in Fig. 8. Updated the code to fix the bug and now the visualization correctly shows that it is the economy of China growing throughout time and not Switzerland.”

10. Figure 7 is not commented at all.

Answer: Our initial approach to the figures was to preview the different kinds of results that can be done. However, according to your comment, we added the following description of Fig. 7:

“Although we are focusing on 6 countries of interest within the paper, on Fig. 7 we can see how the input and output sensitivity rankings for all countries in the WIOD dataset change through time.

We additionally increased the descriptions of Fig. 8-10 by adding the following paragraph:

“The input volatility is noticeably more balanced across the countries compared to output volatility, however, for most countries, the average volatility as an exporter is much higher than the volatility as an importer (Fig. 8). Interestingly, here China has the largest input volatility throughout the entire time period. The USA starts with the largest output sensitivity by far, however, it has a significant drop over the fourteen-year period and is overtaken by the ROW and China economies which unlike most countries achieve a significant growth (Fig. 9). Finally, on Fig. 10 we can see the input and output sensitivity rankings for all countries in the WIOD dataset change through time. While the input volatilities show a high instability through time, this is most likely due to the minute difference within the input sensitivities discussed about previously. On the other hand, output volatilities show the least changes through time, with a few exceptions such as the rise in the output volatility ranking of Russia.”

11. In the appendix, you could add more details to the steps described to derive your model.

Answer: This particular model is quite well-established and has been utilized in various academic works, including those by prominent scholars like Acemoglu. In our study, we extended this model to incorporate the World Input-Output Database (WIOD). However, we didn't emphasize it as a original contribution in our research as it is an elementary extension of the original model. However, we specify the extension nonetheless in Appendix A of the supplementary material. Furthermore, we explain how the theorems can be derived based on this model in Appendix B. We do recognize the importance of clearly elucidating the rationale behind our research choices and are grateful for your feedback on this matter.

12. . Your references in the appendix are not correctly formatted (there are “?” in lieu of all references).

Answer: The appendix is fixed and now correctly corresponds to the references. 

13. The introduction begins with and is substantially (about a third of it) devoted to a discussion on the links between competition, globalization, and international trade dynamics. However, the competition aspects of globalization are not really addressed anywhere in the rest of the paper. It might be better to refocus the introduction on the risks of globalization – i.e., the heart of the model & results. Giving example of these risks would also be beneficial (the COVID crisis is a very obvious one).

Answer: We've rewritten the introduction of the paper to align it more closely with the paper's content and its primary objective: presenting a novel methodology for conducting sensitivity analyses. Furthermore, we've incorporated several references mentioned in response 1, which offer comprehensive insights into the advantages and challenges associated with globalization.

14. You can shorten your description of the WIOD by only retaining the main elements that are useful for your model/analysis. Interested readers can refer back to the database’s documentation to get more information if needed.

Answer: Deleted the following: “In terms of sector classification, ISIC Rev. 4 is similar to the Statistical Classification of Economic Activities in the European Community, commonly referred to as NACE, which is the standard nomenclature of the European Commission for productive economic activities. revision 2 is also based on underlying WIOTs covering 56 sectors.”

15. You don’t describe the second database you use in your “Data” section – the Total Factor Productivity data from the Conference Board Total Economy Database.

Answer: We deleted the part of the Model Implementation section where the database was introduced and added the following subsubsection in the Data section:

“Total Economy Database

To calculate the Hicks-neutral productivity shocks we use the April, 2022 release of the Conference Board Total Economy Database™. We start with a baseline of 100 for each country for the year 1990. Then, using the Total Factor Productivity data we calculate the productivity shocks throughout time for each country. These values are then logarithmically transformed, standardized to a unit variance and normalized to a sum of one due to the model specifications outlined below.”

16. You could explain why you work specifically within the country world-input network, rather than the other 2 possible variations. You choose this variation in particular without really detailing why it is more relevant than the others.

Answer: We randomly choose one network and work on it. However, as explained in the methods, the exact same analysis can be done for any other network. 

17. A(1) is not defined in your main paper - you only define A(2) and A(3) explicitly.

Answer: Changed the introduction of the original A matrix as the A(1) matrix in the World-Input Network:

 “We can now define the World-input network, which is represented by the (J × S) × (J × S) adjacency matrix A(1) = [a(1)_ˆiˆj ] where ˆi and ˆj represent industry pairs (i, r) and (j, s) respectively, such that a(1) ˆiˆj = z^rs_ij /x^s_j”

18. You don’t explain why it is “more reasonable” to use sensitivity as an indicator rather than elasticity in the case of network linkages. Is this something that is standard in the literature? What are the advantages?

Answer: Sensitivity is better suited as it measures the absolute change in the dependent variable in response to a given percentage change in an independent variable. Comparatively, elasticity measures the proportional change in the dependent variable. Due to the additive nature of the Leontief-inverse matrix, we specify that it is “more reasonable” to use sensitivity for this specific scenario. To clarify this within the paper, we rephrase the following sentence:

” In the case of network linkages, it is more reasonable to use sensitivity, due to the fact that we are measuring the absolute change in the Leontief-inverse matrix instead of measuring a proportional change which is more suited for a elasticity analysis”

19. You don’t provide any preview of your qualitative results in the introduction (or in the abstract). You should add highlights of the elements that are most significant and how they relate to existing literature (are they surprising? standard?).

Answer: As our analysis is methodological and our findings are primarily quantitative, we have refrained from drawing direct implications within the abstract. Nevertheless, to underscore the potential relevance of our analytical results for informing future research, particularly in the context of economic analyses and policy implications, we have included the following in the abstract:

“… Our study introduces a novel methodology that enables us to acquire input and output link sensitivities for all country pairings when an economic shock initiates or concludes within a country of interest. This innovative approach also facilitates the analysis of evolving trends in these link sensitivities, providing a comprehensive understanding of the dynamics of shock propagation across the global network…”

20. In your comments on figure 12, you note that there is a “considerable downturn in 2008” but don’t try to explain this. It seems rather counterintuitive as one could think that during the crisis, countries were even more dependent on GVCs, and therefore more sensitive to shocks from other countries. Additionally, you describe the differences between the Z and P tensors, but do not interpret these differences. What do they tell us about overall risk sensitivity in GVC networks?

Answer: As previously emphasized, we aim to maintain a methodological focus in this paper and limit extensive discussions regarding the economic interpretations of our findings. Consequently, our approach will predominantly involve providing a descriptive interpretation of the results. This ensures that our primary objective, presenting and explaining the methodology, remains the central theme of the paper. 

LANGUAGE / MISSPELLINGS

1. P.1, l.4: “pervasive” has a strong negative connotation that does not seem appropriate here 

Answer: Changed to “widespread”.

2. P.2, l.29: “growingly integrated” is clumsy, you might want to change to “increasingly integrated” 

Answer: Changed to “increasingly integrated”.

3. P.2 l.29-32: “GVCs have been used … for International Development.” This sentence needs to be rephrased. 

Answer: Changed to “Major international organizations like the World Bank, the World Trade Organization, the International Labor Organization, and the U.S. Agency for International Development have utilized GVCs in their research and policy development.”

4. P. 2, l.37: “draws attention that” words missing

5. P.2, l. 52: “lenghts" is misspelled

Answer: Changed to “lengths”.

6. P.3, l. 106: you use the WIOT abbreviation without having ever defined it (should be at the beginning of that paragraph when you spell out “World Input-Output Tables”

Answer: Added “WIOT” abbreviation on first use.

7. P.4, l.111: “use” typed twice

 Answer: Deleted second “use”.

8. P.6, l.174: you used the wrong epsilon symbol

Answer: Changed to correct epsilon symbol.

9. P.7, l.203: “the specified axis such the calculation…”: missing “as”

Answer: Changed to “…along the specified axis such as the calculation…”

10. P.7, l.214 + p.8, l.223 : “an J x J matrix” - should be “a J x J matrix”

Answer: Changed to “a JxJ matrix”

---

## [Decision Letter · Decision Letter 1]

16 Oct 2023

Sensitivity analysis of shock distributions in the world economy

PONE-D-22-33523R1

Dear Dr. Domazetoski,

We’re pleased to inform you that your manuscript has been judged scientifically suitable for publication and will be formally accepted for publication once it meets all outstanding technical requirements.

Kind regards,

Emilia Lamonaca

Academic Editor

PLOS ONE

Additional Editor Comments (optional):

Reviewers' comments:

Reviewer's Responses to Questions

**Comments to the Author**

1. If the authors have adequately addressed your comments raised in a previous round of review and you feel that this manuscript is now acceptable for publication, you may indicate that here to bypass the “Comments to the Author” section, enter your conflict of interest statement in the “Confidential to Editor” section, and submit your "Accept" recommendation.

Reviewer #1: All comments have been addressed

2. Is the manuscript technically sound, and do the data support the conclusions?

Reviewer #1: (No Response)

3. Has the statistical analysis been performed appropriately and rigorously? 

Reviewer #1: (No Response)

4. Have the authors made all data underlying the findings in their manuscript fully available?

Reviewer #1: (No Response)

5. Is the manuscript presented in an intelligible fashion and written in standard English?

Reviewer #1: (No Response)

6. Review Comments to the Author

Reviewer #1: (No Response)

7. PLOS authors have the option to publish the peer review history of their article (what does this mean?). If published, this will include your full peer review and any attached files.

Reviewer #1: No

---

## [Editor Report · Acceptance letter]

19 Oct 2023

PONE-D-22-33523R1 

Sensitivity analysis of shock distributions in the world economy 

Dear Dr. Domazetoski:

I'm pleased to inform you that your manuscript has been deemed suitable for publication in PLOS ONE. Congratulations! Your manuscript is now with our production department. 

Kind regards, 

on behalf of

Dr. Emilia Lamonaca 

Academic Editor

PLOS ONE